# A Method for Modeling Acoustic Waves in Moving Subdomains

**Milan Brankovic ***[ID] and **Mark E. Everett** [ID]

Geology and Geophysics, Texas A&M University, 3115 TAMU, College Station, TX 77843-3115, USA; info@geos.tamu.edu

\* Correspondence: mbrankov@tamu.edu or milanovstalni@gmail.com

**Abstract:** Forward modeling plays a key role in both the creation of predictive models and the study of the surrounding environment through inversion methods. Due to their competitive computational cost and modest algorithmic complexity, finite difference methods (FDM) are commonly used to model the acoustic wave equation. An algorithm has been developed to decrease the computational cost of acoustic-wave forward modeling that can be applied to most finite difference methods. An important feature of the algorithm is the calculation, at each time step, of the pressure in only a moving subdomain which contains the grid points across which waves are passing. The computation is skipped at grid points at which the waves are negligibly small or non-existent. The novelty in this work comes from flexibility of the subdomain and its ability to closely follow the developing wavefield. To demonstrate the efficacy of the algorithm, it is applied to a standard finite difference scheme and validated against 2-D modeling results. The algorithm herein can play an important role in the reduction in computation time of seismic data analysis as the volumes of seismic data increase due to developments in data acquisition technology.

**Keywords:** wave propagation; finite difference; forward modeling; seafloor

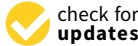

## 1. Introduction

The new simulation method presented in this manuscript is largely motivated by the recent developments in the data acquisition technology. Specifically, it is motivated by the development of distributed acoustic sensing (DAS), which uses fiber optic cables to record both low frequency strain and high frequency seismic waves [1]. DAS provides a new and unique view of the reservoir by sampling cable strain at rapid cadence and at densely spaced locations. High recording rates at numerous receiver positions increase the amount of registered microseismic activity. While it was originally developed for geophysical exploration, DAS has recently seen increasing use in other fields of geophysics. Distributed acoustic sensing has been coupled to existing submarine cables to monitor ground motion signals from seismic events and identify fault zones [2]. Because of its unprecedented spatial and temporal resolutions, DAS is expected to see further use in earthquake monitoring, imaging of faults and other geologic structures, and natural hazard assessments [3]. In conclusion, DAS records data at high frequency and over a long range of densely spaced locations. Furthermore, it can turn fiber-optic cables, which were initially intended for other purposes, into large collectors of seismic data. While DAS is excellent for gathering seismic data, it also has the potential to drastically increase the amount of seismic data recorded in the future.

To prepare for the future increase in the volumes of seismic data, we developed an algorithm to decrease the computational cost of forward wave modeling, which will speed up the processing and analysis of these data. The initial application, presented in this manuscript, is to acoustic waves, modeled by the acoustic wave equation in two dimensional domains, but the algorithm can be extended to three-dimensional models. The appropriate method for modeling waves depends on the purpose of modeling, the size and properties of the modeling domain, and the available computer resources. There does not

exist an ideal finite difference method that can be used in every situation. For example, Zhou et al. [4] show improvements in accuracy via optimization that allows a reduction in the length of the FDM operator. Here, we take an alternate approach for optimizing scalar wave equation simulations. We develop an algorithm that can be used with any finite difference method that utilizes pre-defined finite difference operators and any model discretization regardless of the grid-point distribution. Specifically, the algorithm allows the user to calculate the pressure in only a subset of grid points in the modeling domain through which waves are propagating. Therefore, the numbers of grid points and physical degrees of freedom are reduced, while the grid-point spacing remains the same. Therefore, the numbers of grid points and physical degrees of freedom are reduced, while the grid-point spacing remains the same. However, if the physical nature of the problem is such that active waves are propagating over the entire domain, with no quiet areas, such as in [5], then the RDM method loses its principal advantage.

This is not the first study that aims to speed up a finite difference scheme by modeling waves in only a subset of all the grid points, i.e., in a moving subdomain. Initially, Boore [6] noted that the displacement does not need to be computed in the areas which the first arrival has not yet reached. This idea was further developed when Vidale [7] used an eikonal equation to calculate the arrival times of waves at each grid point and then modeled the evolution of the wave at each grid point for a predetermined amount of time after the arrival. The drawback of this method is that it is focused on modeling only the head waves. There have also been studies published that model the propagation of seismic waves in moving zones (or boxes [8–10]. The path of the box shaped moving zone is pre-defined, and the box represents a subset of the entire modeling domain that is focused on the waves of interest (which are often the head waves). By restricting the modeling to the box enclosing the wave of interest, reflections outside the zone of interest are neglected. In both methods discussed above, the constraints to the modeling subdomain that provide the computational speed-up also restrict the applicability of the method.

In this work we introduce a new flexible approach for selecting the subset of grid points on which the wave is modeled. This method allows for the modeling of reflected waves even if they are far from the first-arriving wavefront. At the neglected, or irrelevant, grid points, disturbances caused by the waves should be small, or even non-existent, depending on the application and user-defined parameters. Because the purpose of our method is to reduce the number of grid points in the domain at which the pressure is calculated, we refer to it as the "reduced domain method" or RDM. By defining certain parameters in RDM, the user may adjust the criterion which differentiates between relevant and irrelevant grid points. Because of this, while the performance may vary, the algorithm can be useful in a large variety of scenarios of wave propagation.

It should also be noted that the most recent application of a moving subdomain, or a moving frame to be more accurate, was for modeling acoustic waves propagating through the earth's atmosphere using the Navier–Stokes equations. The numerical simulations have been performed in two dimensions on Cartesian grids [11], in a two dimensional cylindrical coordinate system with assumed axial symmetry [12,13], and in full three dimensions [14]. While the algorithm developed in our research is implemented for the acoustic wave equation, with additional programming effort it can also be applied to FDMs used for modeling the elastic wave equation as well as the Navier–Stokes equations. This is because the reduced domain method is designed to be applied to any FDM and velocity model, regardless of the grid-point distribution as long as the FDM has pre-defined operators. For simplicity, we will refer to such methods as standard FDMs.

With additional programming effort, the RDM may also be adapted to some methods even if the finite difference operators are not pre-determined. A good example of this is [15], where acoustic waves are modeled while the coefficients in the finite difference operators adaptively change. However, there are some methods such as [16], in which the acoustic waves are modeled by finite difference operators which change length adaptively during the simulation. For such methods, the implementation of RDM becomes more difficult. To

summarize, RDM provides a reduction in computational cost by modeling the waves in only a subset of the entire domain and it can be applied to any velocity model and a variety of FDM. However, thanks to its flexible and adaptive selections of subdomains during the simulation time, it also allows the user to accurately observe the majority of the wavefield. This is an improvement compared to previous methods that modeled waves in moving subdomains.

## 2. Methodology

RDM is a method that first determines the "active" portion of the modeling domain, i.e., the zone within which waves are propagating. The method then uses the selected finite difference method to simulate wave propagation within the active area. Further details are given below.

In finite difference schemes, the pressure field is described by a vector $p$ whose number of elements equals the number of grid points in the modeling domain. We seek to reduce the length of this vector and refer to the new, smaller vector as the "reduced" pressure vector, or simply the reduced vector $p^r$. The elements in the reduced vector at a given time step comprise only the pressure at those grid points through which a wave is actively propagating. We refer to these grid points as "relevant" grid points. As waves propagate through the modeling domain, the set of relevant grid points changes. To find the reduced vector at each time step, RDM determines the set of relevant grid points without actually evaluating the pressure at all the grid points. The following paragraphs and Figure 1 below explain how RDM achieves this goal.

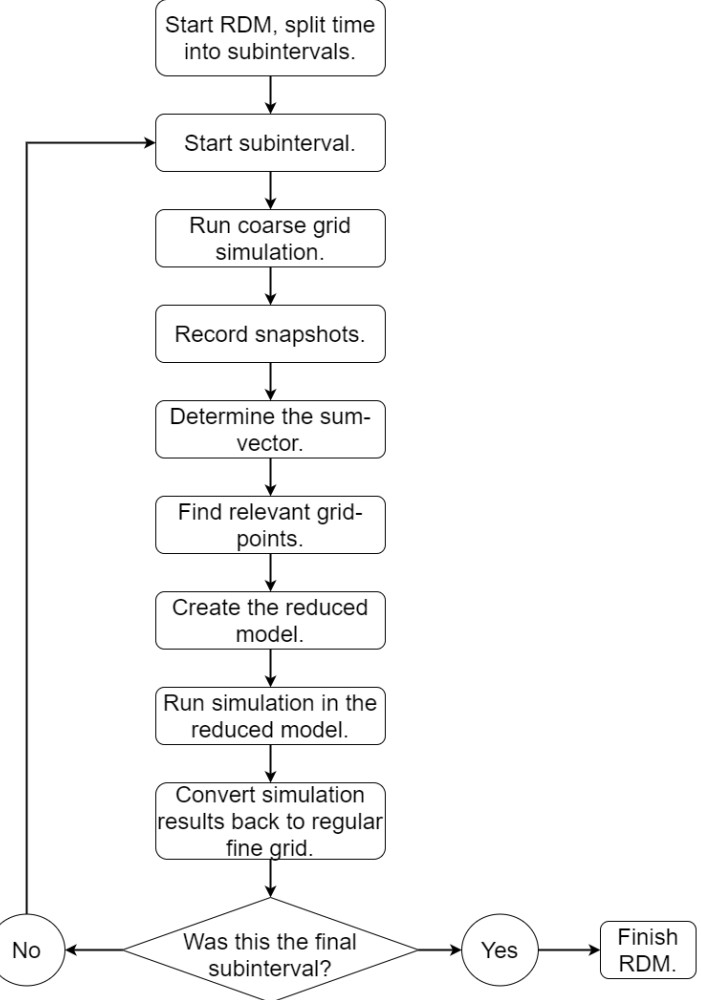

**Figure 1.** A flowchart providing a high-level description of RDM.

The RDM algorithm starts by dividing the time during which the wave propagation is simulated into subintervals of length $T^s$. The subinterval length $T^s$ needs to be small in order to keep the set of relevant grid points within subintervals small. On the other hand, decreasing the subinterval length $T^s$ will increase the number of subintervals in the simulation and the time spent finding the set of relevant grid points for all subintervals could start having a large effect on computation time. The best value for $T^s$ depends on the velocity model and the source, and there does not exist an ideal $T^s$ value that gives the best results in every scenario. However, you can still run consistently fast and accurate simulations while always using the same value of $T^s$. We set $T^s$ to be equal to the period of the dominant frequency of the source, and as can be seen in the following section, RDM drastically reduced computation time and maintained accuracy in all of the tested models.

At the start of each subinterval, prior to advancing the simulation using the reduced vector, RDM runs a fast simulation on a coarse grid that spans the entire modeling domain. The grid-point spacing and the time step of the coarse-grid simulation are set to be twice as large as those in the standard grid, or rather, fine-grid simulation. The FDM that is used to run the simulation in the coarse grid is the same as the FDM used in the fine-grid simulation. For future improvement of RDM one could consider using a different FDM for the coarse-grid simulation, which allows the use of fewer grid points and reduces the computation time. However, on average only 25–30% of RDM computation time is spent in the coarse-grid simulation, so the potential for the computation reduction is limited. The simulation on the coarse grid is not intended to yield a highly accurate displacement field, but it is detailed enough to avoid excessive numerical dispersion and allow a sufficiently accurate determination of the relevant grid points.

The relevant grid points are determined by first defining a "sum vector" $v^{sum}$. Each component of the sum vector is a time summation, over the current subinterval, of the squared time derivatives of the corresponding component of the pressure field on the coarse grid:

$$v_i^{sum} = \sum_{j=1}^{N} \left( \frac{\partial p_i^c(t^0 + (j-1)t^{snp})}{\partial t} \right)^2 \tag{1}$$

where $p^c$ is the pressure vector computed in the coarse-grid simulation, $N$ is the number of snapshots in a subinterval, and $t^{snp}$ is the time between two subsequent snapshots. The magnitude of the sum vector for a given subinterval, at a given grid point, is large if the corresponding pressure on that grid point was large during the subinterval.

The values of $N$ and $t^{snp}$ depend on the length of the subinterval $T^s$ and the time step of the finite difference scheme. Although $N$ and $t^{snp}$ need not have specific values, typically we set $t^{snp}$ such that $40 \geq N \geq 20$. The goal is to use enough snapshots to accurately describe the wavefield during the interval, while also not using so many snapshots as to affect the computation time. It should be noted that changing $N$ and $t^{snp}$ has very little effect on the performance of the simulation. Thus, we do not think that optimizing those parameters can lead to noticeable improvements.

Before the values recorded in $v^{sum}$ are used to estimate the map of relevant grid points, an equal weight averaging filter is applied to $v^{sum}$. Filtering is performed to smooth the results stored in the sum vector, which reduces the length of the bounding curve between the relevant and irrelevant grid points, as can be seen in Figure 2. A big part of the error caused by using RDM is produced at the boundary between relevant and irrelevant grid points. Having wavefield drop from near-zero values to zero can create a source of error in the wavefield. By reducing the length of the boundary between relevant and irrelevant grid points, we reduce the error. This allows us to reduce the computation time more aggressively, while still maintaining a small error. The averaging filter is two dimensional and 32 grid points wide and long. This is because its length is defined as four times the shortest wavelength in the model, i.e., four times the period of the dominant frequency multiplied by the velocity from the slowest area in the model. The processing time of the averaging filter is proportional to the length of the filter and the number of grid points used

in the simulation. As RDM was tested on multiple models, the computation time of the averaging filter varied between 3% and 7% of the entire simulation time when using RDM. To further reduce the computation time of the averaging filter, we could use better picks for the window size that are based on the velocity average rather than minimum velocity, and we could also apply the filter to a different vector that has fewer elements than $v^{sum}$, for example, a vector containing elements of $v^{sum}$ for grid points with spacing four times as large as that of the fine grid. Once the sum vector $v^{sum}$ is determined, the set of relevant grid points is constructed.

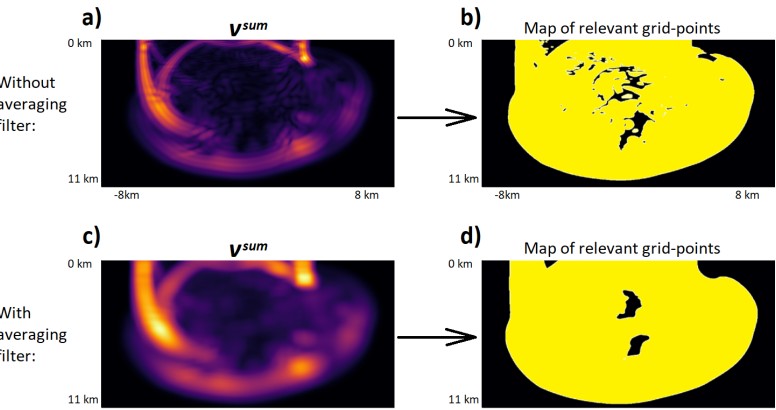

**Figure 2.** (**a**) The sum vector $v^{sum}$ without the averaging filter and (**b**) the resulting map of relevant grid points. (**c**) The sum vector $v^{sum}$ with the averaging filter and (**d**) the resulting map of relevant grid points.

The set of relevant grid points is defined as the smallest subset $U$ of all the grid points such that the sum of elements in the sum vector $v^{sum}$ representing those grid points is greater than or equal to some pre-determined threshold fraction $(1 - e^{-\delta})$ of the sum of all elements in the sum vector:

$$min(n(U)): \quad \sum_{j=1}^{n(U)} v^{sum}_{(U_j)} \geq (1 - e^{-\delta}) \sum_{j=1}^{n(v^{sum})} v^{sum}_j \qquad (2)$$

where $n(U)$ is the number of elements in $U$, $n(v^{sum})$ is the number of elements in $v^{sum}$, and the threshold $(1 - e^{-\delta})$ is defined by the parameter $\delta$. The threshold is defined in this way so that an increase in $\delta$ causes the threshold to increase, converging closer to the value of 1. An increase in the threshold results in more grid points being included in the set of relevant grid points. Therefore, increasing the parameter $\delta$ increases the accuracy of RDM but also increases the computation time.

While the parameters $T^s$, $t^{snp}$ and $N$ should remain the same in all simulations, the parameter $\delta$ can be adjusted to best support the FDM we are applying our method to, which is the purpose of the simulation. For example, if we are doing reverse time migration (RTM) for the purpose of locating a seismic event, we can ignore a lot of weak waves that we know will not contribute to the convergence at the source location. In this case we could set up $\delta$ to a small value that will result in fewer relevant grid points and faster simulation. Alternatively, if the user is interested in simulating weak reflection, the parameter $\delta$ would be set up to a larger value to make sure the weak reflections are represented.

When using Equation (2), RDM determines the set of relevant grid points based on the amplitudes of the propagating waves. This method allows the user to observe the large majority of the wavefield while reducing the computation time. In a more specific example, the user might have a special interest in reflections in a given area of the model, even if the reflections are weak. In such a case, Equation (2) can be modified so that the summations include weights for each grid point. This way, we can add extra importance to grid points

from a specific area. Therefore, depending on the specific purpose of the wave simulations in the future, the parameter $\delta$ and Equation (2) may be adjusted and modified.

Once the set of relevant grid points has been determined, the next step is to create the reduced model. The set of relevant grid points ($U$) is applied to the vectors describing the pressure from the two latest subsequent time steps ($p$, $p^o$) and velocity ($c$) on the fine grid. Specifically, the set of relevant grid points tells us which elements in pressure (or velocity) vectors represent the pressure (or velocity) on the relevant grid points. To generate reduced pressure and velocity vectors, an algorithm goes through all the elements of the pressure vectors $p$, $p^o$, and velocity vector $c$ and the values describing the pressure or velocity at relevant grid points are recorded in reduced pressure and velocity vectors $p^r$, $p^{or}$, and $c^r$. The set of vectors converted to the reduced model may vary between different models and different simulation methods. For example, in cases with heterogeneous density, we also must apply the set of relevant grid points to the density vector ($\rho$) in order to generate the reduced density vector $\rho^r$.

At the next stage of the algorithm, the fine-grid simulation is executed over the subinterval on the set of relevant grid points. The computation is executed on $p^r$ and $p^{or}$ rather than on $p$ and $p^o$, creating a significant reduction in computation time. Once the fine-grid simulation reaches the end of the subinterval, the values of the pressure on the standard fine grid $p$ and $p^o$ are updated using the reduced vectors $p^r$ and $p^{or}$ and the set of relevant grid points $U$. The process is repeated until the simulation reaches the end of the final subinterval.

In the introduction we stated that RDM is used to reduce the cost of modeling the acoustic wave equation. The specific FDM that RDM is applied to is the one by Alford et al. [17], also described in Zakaria et al. [18], which was chosen for its efficiency and simple implementation. Here, the second time derivative of pressure, $\frac{\partial^2 p}{\partial t^2}$, is estimated with a fourth-order accurate nine-point stencil:

$$\frac{\partial^2 p_{i,j}}{\partial t^2} = \Big( (p_{i-1,j} + p_{i+1,j} + p_{i,j-1} + p_{i,j+1}) \frac{4}{3}$$
$$- (p_{i-2,j} + p_{i+2,j} + p_{i,j-2} + p_{i,j+2}) \frac{1}{12}$$
$$- 5 p_{i,j} \Big) \frac{c_{i,j}^2}{\Delta x^2} \tag{3}$$

where $\Delta x$ is the spacing between grid points. Furthermore, the finite difference scheme can be adapted to heterogeneous density models by altering the finite difference coefficients:

$$\frac{\partial^2 p_{i,j}}{\partial t^2} = \Big( \big( (2 - \rho_{i,j}^x) p_{i-1,j} + (2 + \rho_{i,j}^x) p_{i+1,j} + (2 - \rho_{i,j}^y) p_{i,j-1} + (2 + \rho_{i,j}^y) p_{i,j+1} \big) \frac{2}{3}$$
$$- \big( (1 - \rho_{i,j}^x) p_{i-2,j} + (1 + \rho_{i,j}^x) p_{i+2,j} + (1 - \rho_{i,j}^y) p_{i,j-2} + (1 + \rho_{i,j}^y) p_{i,j+2} \big) \frac{1}{12}$$
$$- 5 p_{i,j} \Big) \frac{c_{i,j}^2}{\Delta x^2} \tag{4}$$

where:

$$\rho_{i,j}^x = \rho_{i,j} \Big( \frac{1}{\rho_{i-2,j}} - \frac{8}{\rho_{i-1,j}} + \frac{8}{\rho_{i+1,j}} - \frac{1}{\rho_{i+2,j}} \Big) \frac{1}{12}$$
$$\rho_{i,j}^y = \rho_{i,j} \Big( \frac{1}{\rho_{i,j-2}} - \frac{8}{\rho_{i,j-1}} + \frac{8}{\rho_{i,j+1}} - \frac{1}{\rho_{i,j+2}} \Big) \frac{1}{12}. \tag{5}$$

Equations (4) and (5) above, which are applied to inhomogeneous density models, can be derived from the first equation from [19]. It is important to point out that Equations (3)–(5) do not provide a perfectly accurate representation of the processes in

RDM. Specifically, the pressure, velocity, and density are all recorded as vectors in RDM whereas in Equations (3)–(5) they are presented as matrices. We made this decision because we wanted to provide a more clear and easy-to-read representation of the finite difference operators being used.

In the fine-grid simulations we set the grid-point spacing to be sixteen times smaller than the period of the dominant frequency of the source multiplied by the velocity in the slowest area in the model. This way, there is never fewer than sixteen grid points per wavelength, or eight grid points per wavelength in the coarse-grid simulation. Once the second derivative of the pressure field $p$ is calculated, a second-order accurate scheme uses the current pressure $p$ and the pressure from the previous time step $p^o$ to calculate the pressure field at next time step $p^n$:

$$p^n = 2p - p^o + \Delta t^2 \frac{\partial^2 p_{i,j}}{\partial t^2} \tag{6}$$

where $\Delta t$ is the time-step size of the simulation. To maintain stability of the simulation, the size of the time steps is set to be one half of the grid-point spacing divided by the maximum wave velocity in the model. Therefore, the number of time steps per wave period is dependent on the ratio between the velocities in the slowest and the fastest regions in the model. However, if the model were homogeneous, there would be 32 time steps in the period of the dominant frequency.

## 3. Results

The RDM algorithm was tested on four synthetic models. The first two models comprise scenarios in which a steel object is buried partially or completely beneath the seafloor. They are used to demonstrate the performance of the algorithm and illustrate how the map of relevant grid points progresses along with the waves during a simulation. All calculations in this manuscript are fully 2-D, i.e., an infinite line source excites an infinite structure invariant along strike, the direction parallel to the source. The run time of RDM is compared to the run time of the standard FDM without RDM. The relative error is defined as the change to the final pressure vector resulting from RDM application, specifically:

$$E_{rel} = \frac{||p - p^{RDM}||}{||p||} \tag{7}$$

where $p$ is the pressure vector obtained from standard FDM and $p^{RDM}$ is the pressure vector obtained by applying RDM to the same FDM.

The code was written in Julia programming language, which is designed for rapid execution of numerical simulations. We chose Julia because it can conveniently optimize functions and implement vectorization, thereby reducing computation time. Furthermore, the simulations are executed on a single core of Intel i7-6820HQ which has a base frequency of 2.70 GHz and 16 GB of RAM. Adjusting the code to run on multiple cores would require more programming, but the algorithm is inherently parallelizable.

The first two models are energized by a Ricker source wavelet with peak frequency 2.3 kHz placed 8.4 m above the ocean floor at a midpoint between the two lateral boundaries. In the first model the steel object is not completely buried in the surrounding limestone ocean floor. The velocity and density values used for limestone in this model were obtained from Table 1 of Bayer [20]. In the second model the entire steel object is buried in sediment. The velocity and density values used for sediment were obtained from Hamilton [21]. If a grid point is positioned on the interface of the two layers, an arithmetic average is used to determine the velocity and density at said grid point. These two models are shown in Figure 3, along with the evolution of the map of the relevant grid points throughout the simulation and the final wavefield in each of the two models.

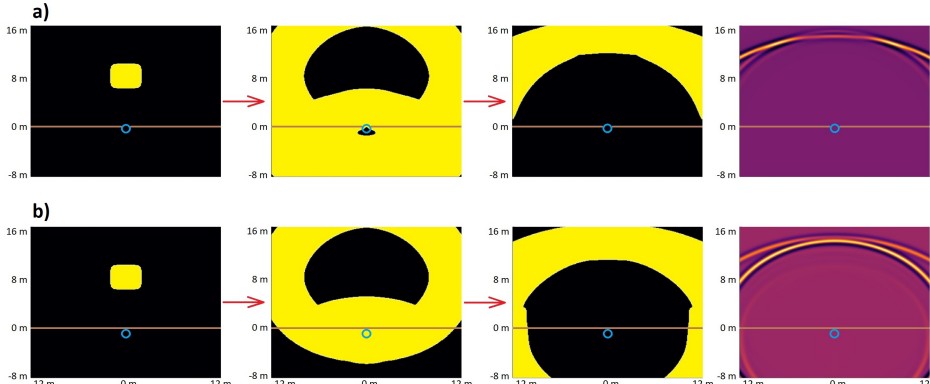

**Figure 3.** (**a**) Results for the first limestone model. (**b**) Results for the second sediment model. From left to right we see: the map of relevant grid points in the first subinterval, nineteenth subinterval, and thirty-seventh subinterval, and finally, the wavefield corresponding to the end of the thirty-seventh subinterval. The figures were acquired during the simulation, with $\delta$ set to 12. The relevant grid points are in the yellow region. The brown line represents the surface of the ocean floor, and the blue line represents the steel object.

**Table 1.** Performance indicators for the first two models.

| | First Model | | Second Model | |
|---|---|---|---|---|
| Parameter $\delta$ | Comp. Time Reduction (%) | Relative Error | Comp. Time Reduction (%) | Relative Error |
| 10 | 56.5 | 0.017 | 63.0 | 0.062 |
| 12 | 54.8 | 0.016 | 61.8 | 0.043 |
| 14 | 48.0 | 0.014 | 56.9 | 0.026 |
| 16 | 40.5 | 0.015 | 49.1 | 0.033 |
| 18 | 40.1 | 0.005 | 50.2 | 0.020 |
| 20 | 40.1 | 0.001 | 47.8 | 0.005 |

In both models, the wave propagation is simulated over a period of 0.016 s, and it took about 255 s with standard FDM to run the simulation, which contained 4515 time steps on a 501 by 501 grid. During the simulations, the waves propagate to the steel object and are reflected past their point of origin. We stated above that the parameter $\delta$ affects the size of the set of relevant grid points, such that an increase in $\delta$ causes an increase in both accuracy and run time. Table 1 displays the computation time reduction and relative error of RDM for various value choices of $\delta$ for the two models.

In the third test we use the velocity model from [22] but assuming a spatially uniform density. The purpose of the modeling is to test RDM on a more heterogeneous velocity model in which many reflections and dispersions are generated. A Ricker source with peak frequency 7.1 Hz is located at the center of the upper boundary of the model domain. The wave is propagated in the simulation for 10.1 s, until it reaches the lateral boundaries of the model domain. The computation time of the standard FDM simulation, which contained 7764 time steps on a 5795 by 1155 grid, was about 9420 s. The velocity model (top), the map of relevant grid points (middle), and the wavefield (bottom) at the end of the simulation are shown in Figure 4. The performance of RDM for different choices of $\delta$ is presented in Table 2.

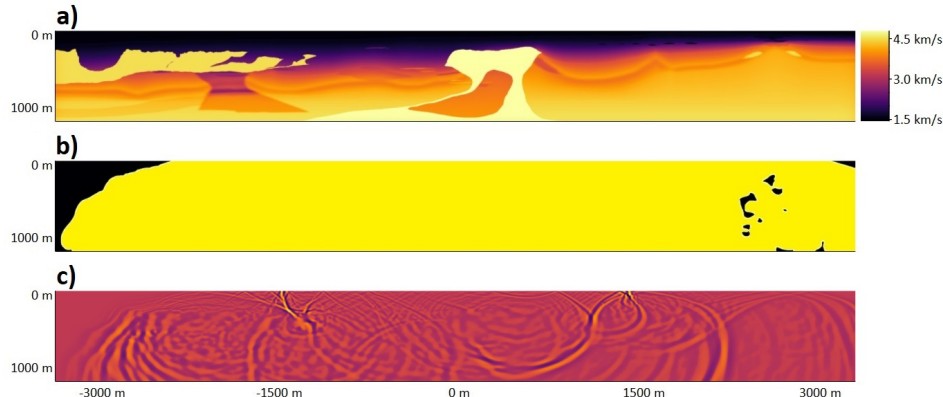

**Figure 4.** (**a**) The velocity model from [22]. (**b**) The map of relevant grid points in the final subinterval. (**c**) The wavefield at the final subinterval of the simulation. The parameter $\delta$ was set to 12.

**Table 2.** Performance indicators for the third model.

| Parameter $\delta$ | Comp. Time Reduction (%) | Relative Error |
|---|---|---|
| 10 | 71.3 | 0.026 |
| 12 | 67.9 | 0.011 |
| 14 | 66.9 | 0.003 |
| 16 | 66.8 | $6.0 \times 10^{-4}$ |
| 18 | 65.9 | $2.7 \times 10^{-4}$ |
| 20 | 65.8 | $6.5 \times 10^{-5}$ |

In the fourth test we combine the velocity and density models from [22] with the same source and simulation time as in the third test. Here, the computation time of the standard FDM simulation, which contained 7764 time steps on a 5795 by 1155 grid was about 10,210 s. The density is strongly heterogeneous which produces a great number of reflections such that every grid point in the model domain is populated with strong coda after the passage of the first-arriving wave. For as long as the strong coda remains, all the grid points through which a head wave has passed would be considered relevant grid points. Thus, we add a new criterion to the selection of grid points that forces RDM to neglect low-amplitude waves. The objective is to assess the accuracy by which RDM models the high-amplitude waves while neglecting weaker ones.

The new criterion is set by fixing the parameter $\delta$ to 20 and adding an extra condition that the number of relevant grid points may not be larger than the total number of grid points multiplied by some fraction $\theta$. This is enforced by adding a new criterion to Equation (2) that states $n(U) \leq \theta n(p)$. This criterion is designed to maintain or ensure a low computation time, wherein the weaker waves are presumably negligible, e.g., below the sensitivity of the recording instruments. The relative error will also be calculated in the zones enclosing the high-amplitude waves, which are presented in Figure 5, along with the density model (top), the map of relevant grid points (middle), and the wavefield (bottom). The performance of RDM on the fourth model is presented in Table 3 below.

**Table 3.** Performance indicators for the fourth model.

| Parameter $\theta$ | Comp. Time Reduction (%) | Relative Error | Relative Error in the Area of Interest |
|---|---|---|---|
| 0.5 | 72.5 | 0.310 | 0.112 |
| 0.6 | 71.7 | 0.229 | 0.073 |
| 0.7 | 71.5 | 0.136 | 0.025 |
| 0.8 | 71.0 | 0.066 | 0.002 |
| 0.9 | 66.7 | 0.005 | $5.7 \times 10^{-7}$ |
| 1.0 | 66.7 | $5.4 \times 10^{-5}$ | $5.7 \times 10^{-7}$ |

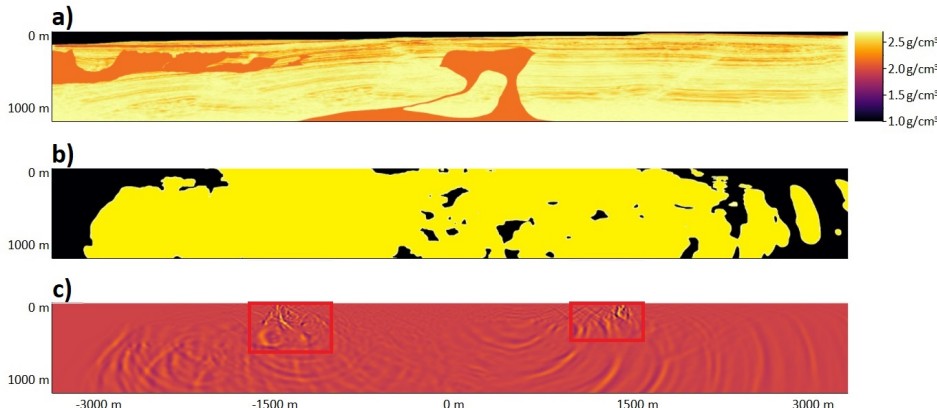

**Figure 5.** (**a**) The density model from [22]. (**b**) The map of relevant grid points at the final subinterval of the simulation with parameter $\theta$ set to 0.7. (**c**) The wavefield at the end of the standard FDM simulation with the strong waves marked with red squares.

## 4. Discussion

The data in Tables 1 and 2 show that the reduction in computation time from RDM ranges between 40 and 70%, depending on the modeling scenario and the value of the $\delta$ parameter. Even though the third model is more heterogeneous than the first two, the reduction in computation time is greater in the third test. This is because the first two modeling domains are much smaller, so the waves reach the boundaries earlier, and therefore the map of relevant grid points expands across the domain more rapidly than in the third model.

While relative error maintains low values in the first three tests, from several percent to $10^{-4}$, in the fourth test it reaches 0.3. The large relative error in the fourth test occurs because an aggressive criterion is used to select relevant grid points, so that many of the weaker waves are not modeled. The goal in the fourth test is to efficiently yet accurately model the high-amplitude waves. This goal is achieved as the relative error in areas enclosing the strong waves (presented in Figure 5) is much smaller, as shown in Table 3.

As stated in the introduction, RDM can be applied to any FDM with pre-defined spatial operators. This means that RDM can also be applied to other, higher-order FDM. Higher-order FDM are generally used to reduce the number of grid points needed in the model, which reduces the memory requirements and can also lead to lower computation times. We expect the percent reduction in computation to remain the same as RDM is applied to various FDM. This is because we do not expect that changing the FDM to which RDM is applied would have a noticeable effect on the RDM's estimation of what portion of the domain contains irrelevant grid points. As long as there are parts of the domain with weak or non-existent waves, RDM can identify areas with irrelevant grid points in the simulation, and the computation time can be reduced. However, if the total number of grid points in the domain decreases as a result of the use of higher-order FDM, the perimeter of the areas of relevant and irrelevant grid points would become more coarse. This could cause the error due to RDM application to increase slightly. It should also be noted that the application of RDM does not create any new limitations on the frequency range of waves that can be modeled in the simulation. The frequency range is entirely dependent on the grid-point spacing, time-step size, and the FDM to which RDM is being applied.

The calculations in this manuscript are performed in 2-D models. The simulations can therefore be used to describe a line source and the response from a structure invariant along strike, which is the direction parallel to the source. In such a scenario, the line of receivers, possibly provided by DAS, can be oriented in any direction relative to the source.

There are several possibilities for future work. The RDM method can be adapted to more complex finite difference schemes or to 3-D applications. Currently, we are more focused on developing a 3-D version of RDM as it will have a greater impact on the range of applications of RDM. This will take time and additional programming effort, but we

expect RDM to be able to maintain its significant reduction in computational cost in the three-dimensional simulations.

## 5. Conclusions

Wave modeling methods that allow the calculation of pressure in only a subset of grid points can provide an excellent reduction in computation time and complement a variety of finite difference schemes. The RDM algorithm developed herein can be applied to any FDM that uses pre-defined finite difference operators. The developed method uses an adaptively changing subset of all the grid points to accurately model both first-arriving and reflected waves. As a result, a high level of accuracy is obtained while reducing computation time by more than 50%. The reduced domain method was tested on simple models describing the ocean floor with a buried steel object, which were used to demonstrate how the map of relevant grid points changes throughout the simulation, and also on more complex and realistic models which contained many layers and substantial heterogeneity.

While RDM is valuable in most forward modeling scenarios, we expect RDM to be the most useful in inverse problems wherein many forward modeling runs are required. Running repeated forward models can take a lot of time. Here, the computation cost reduction to forward modeling obtained with RDM could be of great value. Furthermore, RDM may also be very useful when studying a seismic source with reverse time migration (RTM), where we are only concerned with the waves converging at the source. Because a big portion of the waves in RTM does not converge at the source, we could apply RTM aggressively to drastically reduce computation time, while having little effect on the accuracy of the results.

**Author Contributions:** Conceptualization, M.B.; Methodology, M.B.; Software, M.B.; Supervision, M.E.E.; Validation, M.B. and M.E.E. All authors have read and agreed to the published version of the manuscript.

**Funding:** This research received no external funding.

**Institutional Review Board Statement:** Not applicable.

**Informed Consent Statement:** Not applicable.

**Data Availability Statement:** The velocity models used can be found in references and the code used for running RDM can be accessed by contacting the corresponding author.

**Acknowledgments:** We would like to thank Richard Gibson for insightful discussions.

**Conflicts of Interest:** The authors declare no conflict of interest.

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
