# Peer review of "A Method for Modeling Acoustic Waves in Moving Subdomains"

_acoustics, doi:10.3390/acoustics4020024_

Round 1

Reviewer 1 Report

Comments in the attached pdf

Reviewer 2 Report

This manuscript presents a novel and flexible method for modeling acoustic waves by selecting the subset of grid points. The authors demonstrated that this method reduces computation time and improves the accuracy compared to the standard finite difference method.

This paper was written clearly in a well-structured manner. The presented modeling results are scientifically sound and appropriate to illustrate the difference from the standard method. The conclusions are consistent with the evidence and arguments presented. 

Here are some of my questions for the authors:

  1. It's good to see that the presented method was compared to previous studies, but the cited references [4-8] are kind of outdated. Is there any recent ( within the last 5 years) reference that can be cited?
  2. The efficiency of the RDM method was illustrated by the computation time reduction in percentage. What's the general computation time for completing one simulation? How much RAM did the authors use? Is there any limitation on the frequency of waves that can be modeled by this method? 

Reviewer 3 Report

I have attached file to this email

Round 2

Reviewer 3 Report

I have attached file to this email

Author Response

Thank you very much for your comment. We’ve incorporated the first two references into their appropriate places within the manuscript as a means of updating the reference list.